



# The CopterSonde: An Insight into the Development of a Smart UAS for Atmospheric Boundary Layer Research

Antonio R. Segales[1,2,3], Brian R. Greene[2,3,4], Tyler M. Bell[3,4], William Doyle[3], Joshua J. Martin[3], Elizabeth A. Pillar-Little[3], and Phillip B. Chilson[2,3,4]

[1]University of Oklahoma School of Electrical and Computer Engineering, Norman, Oklahoma
[2]Advanced Radar Research Center, University of Oklahoma, Norman, Oklahoma
[3]Center for Autonomous Sensing and Sampling, University of Oklahoma, Norman, Oklahoma
[4]University of Oklahoma School of Meteorology, Norman, Oklahoma

**Correspondence:** A. Segales (tony.segales@ou.edu)

**Abstract.** The CopterSonde is an uncrewed aircraft system developed in-house by a team of engineers and meteorologists at the University of Oklahoma. The CopterSonde is an ambitious attempt by the Center for Autonomous Sensing and Sampling to address the challenge of filling the observational gap present in the lower atmosphere among the currently used meteorological instruments such as towers and radiosondes. The CopterSonde is a unique and highly flexible platform for in situ atmospheric

boundary layer measurements with high spatial and temporal resolution, suitable for meteorological applications and research. Custom autopilot algorithms and hardware features were developed as solutions to problems identified throughout several field experiments carried out since 2017. In these field experiments, the CopterSonde has been proved capable of safely operating at wind speeds up to $22\,\mathrm{m\,s^{-1}}$, flying at $3050\,\mathrm{m}$ above mean sea level, and operating in extreme temperatures: nearly $-20\,^{\circ}\mathrm{C}$ in Finland and $40\,^{\circ}\mathrm{C}$ in Oklahoma, United States. Leveraging the open-source ArduPilot autopilot code has allowed for seamless

integration of custom functions and protocols for the acquisition, storage, and distribution of atmospheric data along with the flight control data. This led to the creation of features such as the "wind vane mode" algorithm which commands the CopterSonde to always face into the wind. It also allowed for the design of an asymmetric airframe for the CopterSonde, which is shown to have more convenient locations for weather sensor placement, in addition to allowing for improvements in the overall aerodynamic characteristics of the CopterSonde. Moreover, it has also allowed the team to design and create a modular

shell where the sensor package is attached and which can run independently of the CopterSonde's main body. The CopterSonde is on the trend towards a smart UAS tool with a wide possibility of creating new adaptive and optimized atmospheric sampling strategies.



# 1 Introduction

The atmospheric boundary layer (ABL) is a dynamic system that experiences significant changes in the thermodynamic and kinematic states in its vertical and horizontal structure. An understanding of these structures is key for improving numerical modeling, simulations, and weather forecasts. A combination of fine-scale domain models along with higher resolution observations in space and time are required in order to achieve such understanding. The measurement of temperature, humidity, pressure and winds are some of the most important parameters for the description of the thermodynamic and kinematic behav-

ior of the atmospheric boundary layer. There are currently several meteorological instruments able to measure these parameters effectively; however, they are limited in coverage and have high operating costs.

Uncrewed aircraft systems (UAS) are an emerging technology with a growing interest for weather research and atmospheric monitoring in the scientific community. Correspondingly, we are witnessing rapid developments in the autopilot capabilities, ground station software, and airframes. However, a well proven design that fully satisfies the requirements of measuring the

atmospheric parameters accurately and effectively is still in its early stages of development. The first attempts of using UAS for atmospheric research go back to the 1970s (Konrad et al., 1970) and, since then, the integration of sensors aboard the system as a whole has been gradually improving. For instance, the Small Unmanned Meteorological Observer (SUMO) was developed at the University of Bergen as a cost-effective atmospheric measurement system and it is based on a fixed-wing aircraft (Reuder et al., 2009). Later, Wainwright et al. (2015) and Bonin et al. (2015) showed clear examples of the use of the

fixed-wing SUMO for estimating the temperature structure function and compared them against Large-Eddy simulations and sodar observations respectively. Furthermore, the multipurpose airborne sensor carrier (MASC) UAS was recently developed by the Eberhard–Karls University. It was originally designed for boundary layer research, capable of collecting temperature, humidity and wind data in situ at high resolution (Wildmann et al., 2014).

Nowadays, the development of a fully autonomous system that can operate with little to no human intervention seems to be

the next big step in the field. However, the airspace over the United States is regulated by the Federal Aviation Administration (FAA), and their main mission is to deconflict the airspace and keep the airways safe. Therefore, the FAA rules must be complied with since the UAS is an aircraft capable of flying within such airspace. At the time this paper was written, the FAA requires the UAS operator to be physically present in the flight site and it also explicitly states that the UAS must be within sight at all times, unless special permissions from the FAA were acquired. Hence, the current UAS operations are limited by

legal factors. However, there are strong interests from the scientific community in collaborating with the FAA and providing safe solutions with risk mitigation. Chilson et al. (2019) envisions the future concept of operations of a large network of fully autonomous and unattended UAS distributed over the state of Oklahoma and beyond, also known as the 3D Mesonet concept.

The CopterSonde rotary-wing UAS is part of this larger project, and it was specifically designed to be a reusable and safe system capable of scaling and adjusting to the current and future rules and conditions. Moreover, the CopterSonde was developed

as a flexible and cost–effective platform (below 5000 US dollars in materials) for the precise measurement of temperature, humidity, pressure and wind profiles. Additionally, the CopterSonde will receive firmware improvements over time aimed towards a more autonomous and unattended system. In this paper, the design process and the technical capabilities of the CopterSonde





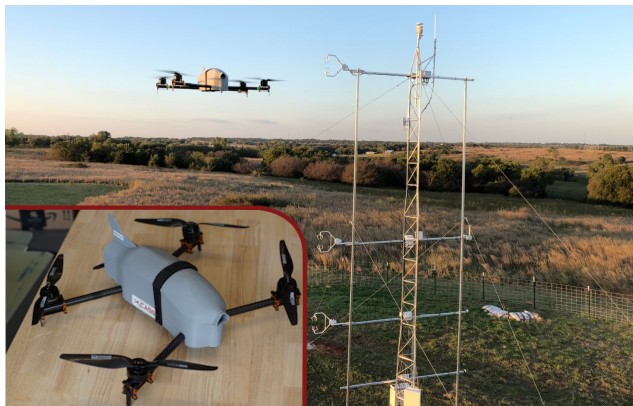

**Figure 1.** Picture of the CopterSonde UAS hovering next to a 10 m experimental flux tower. The picture in the bottom left corner shows a close up of the fully assembled CopterSonde UAS.

are discussed. Results from some field campaigns are also shown to demonstrate the ability of the CopterSonde to repeatably and consistently collect accurate data while sometimes having to endure challenging conditions such as low-level jet winds, icing events, and low air density at high altitudes.

## 2 The CopterSonde system

The Center for Autonomous Sensing and Sampling (CASS) from the University of Oklahoma (OU) made possible the development of the CopterSonde UAS for weather research, shown in Fig. 1. The foundation of the CopterSonde design started with a commercially available airframe equipped with an open–source autopilot system. In the most recent version, the airframe and autopilot have gone through heavy modifications in order to optimize it for atmospheric data acquisition, in particular for temperature, humidity, pressure, and wind data.

### 2.1 Airframe

The presented CopterSonde airframe is based on a modified version of the Lynxmotion HQuad500 construction frame kit from RobotShop Distribution Inc. The CopterSonde is a rotary-wing vehicle that has four fixed pitch rotors mounted at the end of four almost equally spaced arms attached to the main body, also known as quadcopter. The structure of the quadcopter is made of G10 fiberglass plates, carbon fiber tubes, aluminum tube-clamps and stand-offs. A combination of T-motor U3 700 KV motors with carbon fiber T-style propellers 11" × 5.5" (27.94 cm×13.97 cm) are attached at the end of each arm. The motors are controlled by a Lumenier 30A BLHeli32 4-in-1 speed controller which are powered by a single 4 cell 6750 mAh lithium polymer battery. The battery has its own compartment inside the CopterSonde to avoid exposing it to extreme environments and also covered by the shell to protect it from external hazards. The CopterSonde technical and performance specifications are summarized in Table 1.



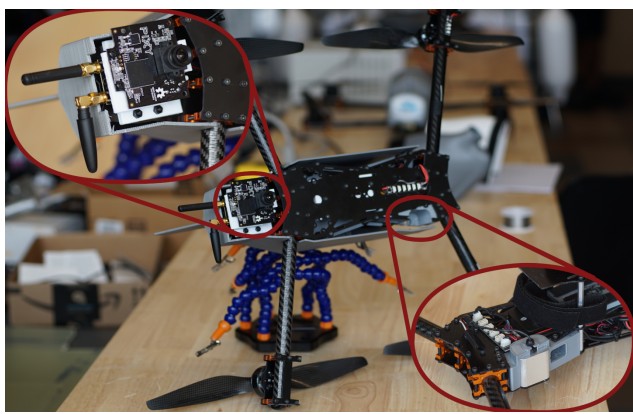

**Figure 2.** Bottom view picture of the CopterSonde UAS illustrating the location of the custom mounts for the LightWare 20/C lidar type rangefinder (bottom right corner) and the IR-LOCK precision landing camera (top left corner).

The original frame was intended for simple recreational flights, in particular for long endurance first person view (FPV) flights. Therefore, the quadcopter had to undergo some modifications to accommodate desired components and repurpose the functionality of the aircraft for weather data collection. Additional components were fitted in the CopterSonde, which included a high precision Here+ V2 Ublox M8P GPS that has the ability to process real-time kinematic (RTK) GPS positioning with a precision on the order of few centimeters. Also, a lidar based rangefinder was mounted on the CopterSonde to accurately measure the height close to the ground which works in conjunction with a precision landing system based on an infrared camera. Figure 2 shows the 3D printed mount supports for the lidar device and the precision landing camera respectively, which were integrated into the structure of the quadcopter. The 3D printed parts, including the shell that encloses the body of the CopterSonde, were modeled using the SolidWorks® computer aided design (CAD) program. Several iterations of the different CAD parts were printed with Polylactic Acid (PLA) plastic using a LulzBot® TAZ 6 3D printer until a perfect fit was attained. Furthermore, the digital models were also useful to study the airflow behavior and its trajectory around such parts and the CopterSonde by using computational fluid dynamic (CFD) simulations, which is discussed in Sect. 2.5.

The arrangement of the electronic components within the CopterSonde was carefully planned to increase its modularity and facilitate performing routine maintenance. In particular, the payload was strategically placed at the frontmost section of the CopterSonde. Consequently, the sensors are subjected to a cleaner airflow by keeping the payload always facing into the wind. As a result, undesired data contamination (produced by sources such as the heat emanated by the CopterSonde itself) is significantly reduced. This technique is better described in Sect. 2.3.

## 2.2 Autopilot Software System

The CopterSonde is equipped with a Pixhawk Cube 2.1 autopilot board (Hex Technology, Sha Tin, Hong Kong), which is used as the main controller for flight stabilization and navigation. The Pixhawk is an open-hardware project that provides users with readily available, low cost and high-end autopilot board solutions for academic, hobby, and industrial communities. It contains a



**Table 1.** Technical specifications of the CopterSonde airframe.

| | |
|---|---|
| Frame size | 500 mm |
| All-up weight | 2.25 – 2.36 kg |
| Maximum speed | 26.4 m s$^{-1}$ |
| Maximum ascent rate | 12.2 m s$^{-1}$ |
| Maximum descent rate | 6.5 m s$^{-1}$ |
| Maximum altitude above ground$^a$ | 1800 m |
| Maximum altitude above sea level | 3050 m |
| Maximum wind speed tolerance | 22 m s$^{-1}$ |
| Flight endurance$^a$ | 18.5 min |
| Operating temperatures$^b$ | $-20\,°C – 40\,°C$ |

$^a$ under favorable weather conditions with almost no wind.

$^b$ tested temperatures, the range can be larger than stated.

powerful microcontroller capable of executing complex autonomous missions. The Pixhawk board runs the ArduPilot autopilot code, which is a free software package that can be redistributed and modified under the terms of the GNU General Public

License version 3 (GPLv3). This means that anyone is free to use all of the code and tools provided in the ArduPilot Github repository without authorization or involvement from the ArduPilot team. However, it is the responsibility of anyone using the code to inform the end user that an open-software is used and the source must be provided. Therefore, under the given license terms, the code source release made by CASS is available to the users in Segales et al. (2019), which is in accordance with the ArduPilot's GPLv3 licence.

The motivation behind the decision taken to modify an existing autopilot code was the flexibility of incorporating additional desired features not dependent on proprietary codes from companies, which usually do not follow the same research line and are prone to being discontinued eventually. By having full access to the UAS's autopilot code and being able to modify it as desired, it allowed for considerable reduction in the electronic hardware by centralizing functions in a single processing unit. For instance, separate data loggers for the weather sensors were not needed since the Pixhawk autopilot board has sufficient

computing power to accommodate the extra data. This also came with the benefit of weight reduction which is crucial to achieve good performance and long endurance during flights. The official ArduPilot code by default does not support the desired meteorological sensors nor the adaptive flight behaviors for optimal atmospheric data acquisition. Therefore, the ArduPilot code has undergone modifications to incorporate custom user code functions and adapt it to the specific application.

In the current version of the CASS–ArduPilot code run by the CopterSonde, a set of custom functions were added on top of

the original ArduPilot code. These are itemized and described as follows.

1. Weather sensor integration: code libraries to read PT-100 thermistors distributed by International Met Systems (iMet) and HYT-271 humidity sensors distributed by Innovative Sensor Technology (IST) were added into the autopilot code.





The communication between the sensors and the autopilot is done over a single bus using the I$^2$C protocol, capable of sampling and storing up to 8 sensors at 20 Hz each to an internal SD card.

2.  Custom sensor message for wireless streaming: since the sensors' data are supplied to the autopilot board, these were then added to the data stream transmitted from the CopterSonde to the ground control station (GCS). Consequently, the sensors can be monitored in real-time while in flight. Additionally, a copy of the streamed data are stored in the GCS's computer.

3.  Wind vane mode: a custom wind estimation algorithm was developed and implemented. The autopilot estimates the
wind direction and adaptively turns the CopterSonde into the wind. By maintaining the CopterSonde orientation into the oncoming wind, the air being drawn across the sensors has not been disturbed by effects from the CopterSonde body. As a result, data contamination is minimized which is crucial when measuring small fluctuations of temperature and humidity, as shown in Greene et al. (2019).

4.  Smart fan for sensor aspiration: the chosen weather sensors have a delicate structure and it must be kept protected from
dust and small debris. Therefore, the sensors were mounted inside a tubular shield and aspirated by a smart fan, which is controlled by an algorithm executed by the autopilot. It toggles the fan's power on/off at specified heights after takeoff and before landing.

Before flashing the custom code into the Pixhawk autopilot board, a thorough debugging of the code can be achieved by using a simulator included in the ArduPilot code. This prevents incidents or malfunctions when running the code for the first time
during a real flight and it also significantly reduces the software development time. Figure 3 shows a screen capture of the simulator's graphical user interface with a temperature and humidity live data monitor developed in house. Moreover, the simulator can create different environmental scenarios, such as adding wind and turbulence, for further evaluations. Additionally, given that the ArduPilot repository is managed by a code hosting platform for version control and collaboration, known as Github, then the update releases from the official ArduPilot account can be easily tracked and merged to the custom forked code. This
feature allows developers to continue harnessing the power of the ArduPilot code for future integrations and optimizations for any desired application.

## 2.3 Adaptive Sampling: The Wind Vane Mode

The approach used to estimate the 2D wind vector on the CopterSonde is based on the measurement data of its onboard inertial measurement unit (IMU). The advantage is that there is no need for additional dedicated airspeed sensor or anemometer,
which is challenging to implement properly on a multicopter and would have also reduce the available space and weight limit for other valuable payload components. The approach presented the challenging task of accurately measuring the 2D wind vector because of the perturbations caused by the propeller wash of the rotors. This disturbs the aerodynamic field around the airframe of the CopterSonde making it difficult to measure small wind fluctuations. As a good practice, the algorithm was designed to be executed on the Pixhawk autopilot board with as few math operations as possible to minimize computing





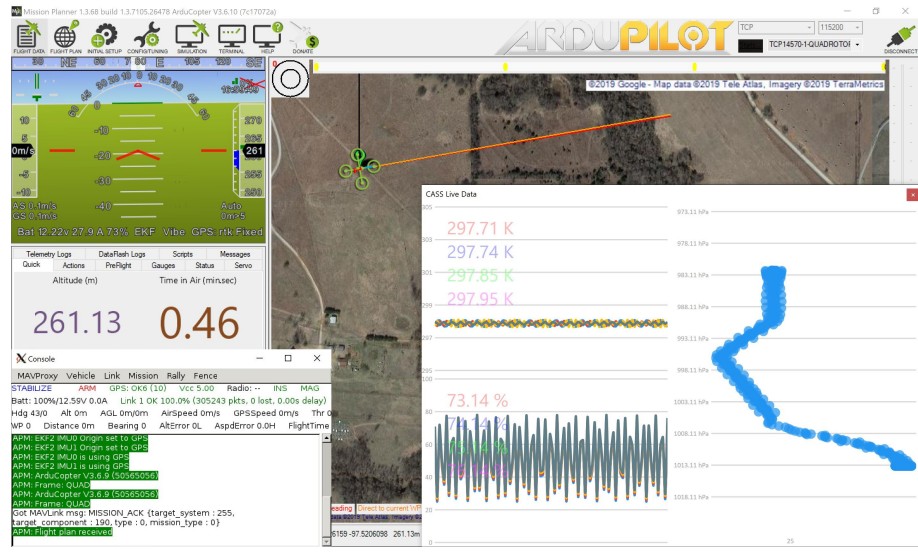

**Figure 3.** Screen capture of the simulator's graphical user interface in which every flight parameter is displayed as if flying a real UAS. The UAS is represented by the green quadcopter in the map provided by Google (©2019 Google - Map data: ©2019 Tele Altas, Imagery ©2019 TerraMetrics), the heads up display (HUD) in the top left corner shows the necesary avionics for the operator. The inset picture in the bottom right corner is a live view of the simulated temperature and humidity data interacting with the simulated UAS. Work is underway to better represent the vertical temperature and humidity profiles in the simulator.

time. Therefore, the dynamic equations were not considered, and only the kinematic equations of motion of the CopterSonde were used in the algorithm. Figure 4 shows the reference frame convention used for vector calculations, where $\boldsymbol{n}_{yz}$ is a unit vector in the direction of X normal to the plane YZ of the inertial frame $I : \{X, Y, Z\}$, $\theta$ and $\phi$ are pitch and roll angles respectively, and $\boldsymbol{e}_\theta$ and $\boldsymbol{e}_\phi$ are unit vectors of the body reference frame. The angle between the vector $\boldsymbol{v} = (\boldsymbol{e}_\theta \times \boldsymbol{e}_\phi)_{xy}$, normal to the CopterSonde's horizontal plane, and the axis $Z$ is a measure of the tilt of the CopterSonde and it is denoted by

$\gamma$. The magnitude and direction of the projection of $\boldsymbol{v}$ onto the plane XY of $I$ can be directly associated with the wind speed and direction respectively. Subsequently, the wind direction with respect to the CopterSonde's true heading ($X$ axis) is

$$\lambda = \arccos\left(\frac{\boldsymbol{n}_{yz} \cdot (\boldsymbol{e}_\theta \times \boldsymbol{e}_\phi)_{xy}}{|\boldsymbol{n}_{yz}| \cdot |(\boldsymbol{e}_\theta \times \boldsymbol{e}_\phi)_{xy}|}\right), \tag{1}$$

and after some vector operations and math simplifications, $\lambda$ results in

$$\lambda = \arctan\left(\frac{-\cos\theta\sin\phi}{\sin\theta\cos\phi}\right), \tag{2}$$

where

$$\boldsymbol{e}_\theta = \begin{bmatrix} \cos\theta \\ 0 \\ -\sin\theta \end{bmatrix} \text{ and } \boldsymbol{e}_\phi = \begin{bmatrix} 0 \\ \cos\phi \\ \sin\phi \end{bmatrix},$$



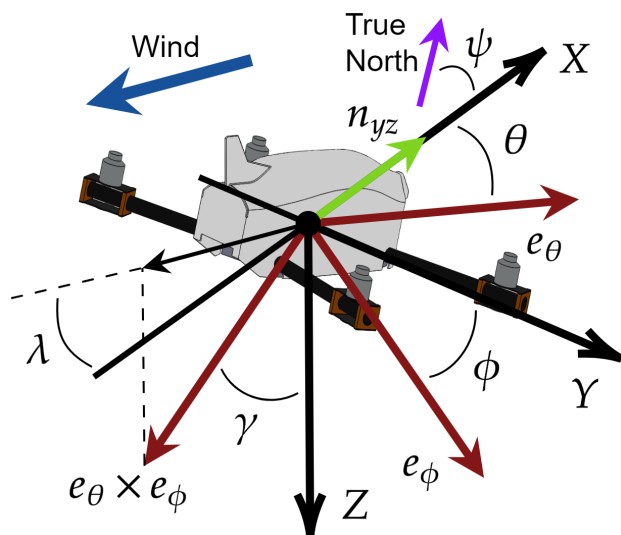

**Figure 4.** Reference frame convention used by ArduPilot and illustrated with the set of orthonormal vectors $I : \{X, Y, Z\}$. The plane described by $X$ and $Y$ is parallel to the surface of the Earth, while the axis $Z$ points downwards to the center of the Earth. The horizontal plane fixed to the CopterSonde body is described by $e_\theta$ and $e_\phi$, and their cross product $e_\theta \times e_\phi$ is the tilt vector used for calculating the wind.

as similarly deduced by Neumann and Bartholmai (2015). Additionally, the CopterSonde is constantly measuring its heading with respect to the true north by means of an onboard compass which is denoted by $\psi$. The sum of $\psi$ with the computed angle $\lambda$ gives the absolute wind direction. However, since the algorithm commands the CopterSonde to turn into the wind, then the angle $\lambda$ tends to zero and, consequently, $\psi$ tends to the absolute wind direction. It should be noted that the wind vane algorithm works only when the CopterSonde is horizontally steady, and the tilt angle $\gamma$ is produced only as a result of the CopterSonde compensating for wind.

The autopilot board executes the wind vane code and calculates the wind direction $\lambda$ at a frequency rate of 10 Hz. Despite the fast computation of $\lambda$, it is better to take the average of a sequence of $\lambda$s to filter out any perturbations caused by the prop wash and other undesired aerodynamic disturbances. The time-averaged angle $\overline{\lambda}$ is then inserted directly into the autopilot flight control algorithm as a yaw command, turning the CopterSonde about its yaw axis to face it into the wind. After several bench tests of the code with the simulator, the time interval for the average was set to $5\,\mathrm{sec}$ or $50$ samples. It was found that a shorter time interval causes the CopterSonde to oscillate about its yaw axis, while a longer time interval outputs wind direction estimates at a very slow rate.

## 2.4 Shell and Payload

The CopterSonde was designed to be a modular system, in particular the payload has its own detachable compartment capable of operating independently from the main body of the CopterSonde. This feature is particularly useful for calibration and maintenance purposes. The CopterSonde's shell is divided into two pieces: the front shell and the back shell, as shown in





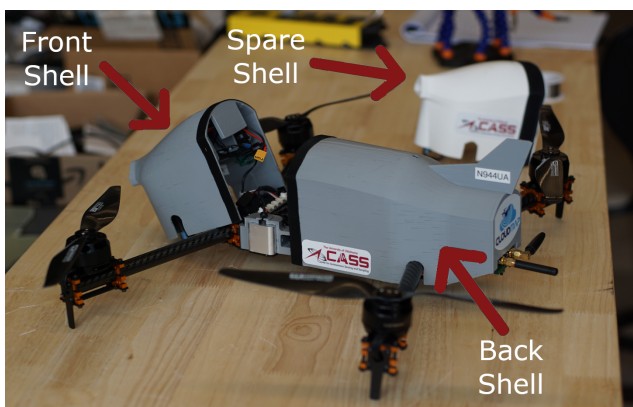

**Figure 5.** Side view picture of the CopterSonde UAS illustrating the modular parts of the 3D printed shell. The front shell is shown unmounted and a spare front shell is shown in the background to exhibit the ease of interchangeability of the CopterSonde's shell.

Fig. 5. The back shell's main purpose is to protect the vital components inside the CopterSonde such as the battery and the autopilot. Moreover, it makes the CopterSonde more aerodynamic and helps orienting the CopterSonde into the wind with its rear vertical fin. The sensor compartment is located in the front shell of the CopterSonde. The shells are exchangeable, which provides the option to have spare shells with different sensor configurations for a broader range of experiments. The dimensions of the compartment are approximately $100 \times 95 \times 70$ mm and can carry a maximum payload weight of $205$ g and a minimum of $90$ g (not including the shell itself). The CopterSonde's center of gravity (CG) without the shells is slightly shifted to the back with respect to its center. Under minimum payload weight configuration, the CG is located at the center of the CopterSonde which is optimal. Conversely, the CG is slightly forward under maximum payload weight configuration without compromising the CopterSonde's stability. The current version of the CopterSonde's front shell is outfitted with a set of three thermistors and three humidity sensors for redundant thermodynamic sampling. Figure 6 shows the location of the sensors and the fan inside the L–shaped duct incorporated into the front shell. The sensors were placed in an inverted "V" shape pattern to prevent the sensors from sampling heated and disturbed air produced by the wake and self-heating of the upstream sensors. The fan was calibrated to draw air across the sensors at a constant speed of $12$ m s$^{-1}$. It automatically switches on (off) after takeoff (before landing) to protect the sensors from dust close to the ground. Further analysis and considerations about the sensor placement on the CopterSonde can be found in Greene et al. (2018) and Greene et al. (2019).

The chosen temperature and humidity sensors are the iMet-XF PT-100 and HYT-271 respectively, because of their small size, low weight, low cost and ease of programming. Pressure is measured by a single MS5611 sensor built-in the Pixhawk autopilot board which is also used for altitude control. The specifications of the sensors are summarized in Table 2. The temperature and humidity sensors were programmed to use the I$^2$C communication protocol to exchange data streams with the Pixhawk autopilot board. Only two connecting cables are required to transfer data between the payload and the CopterSonde's autopilot board. Additionally, the shell carries its own internal battery that powers the sensors and the fan. It can last up to 4 consecutive flights before needing to be replaced. It should also be mentioned that the front shell can be redesigned to fit other types of





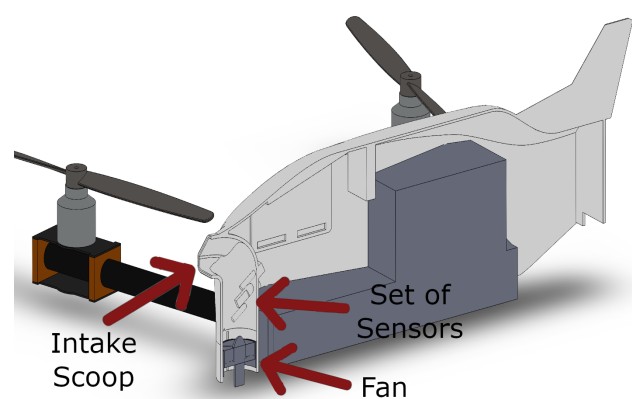

**Figure 6.** Longitudinal section cut view of the CopterSonde CAD model showing the different parts of the inside of the L–shaped duct. The sensors were placed as far from the intake as possible to reduce the heating produced by sun radiation. The airflow induced by the fan circulates from top to bottom at a constant speed.

**Table 2.** Technical specifications of the weather sensors mounted on the CopterSonde.

| Sensor | iMet-XF PT-100 | HYT-271 | MS5611 |
|---|---|---|---|
| Type | Temperature | Humidity | Absolute pressure |
| Sensing element | Bead thermistor | Capacitor | MEMS[a] |
| Range | $-90$–$50\,°C$ | $0$–$100\,\%$ | $10$–$1200$ mbar |
| Response time[b] | $\leq 2$ sec | $\leq 5$ sec | $\leq 8.22$ msec |
| Resolution | $0.01\,°C$ | $0.1\,\%$ | $0.065$–$0.012$ mbar |
| Accuracy[c] | $\pm 0.3\,°C$ | $\pm 0.1\,\%$ | $\pm 1.5$ mbar |
| Sampling rate | 20 Hz | 20 Hz | 20 Hz |
| Protocol | I$^2$C | I$^2$C | SPI |

[a] Micro Electro–Mechanical System.

[b] at $5\,\mathrm{m\,s^{-1}}$ airflow across.

[c] at 25 °C ambient temperature.

sensors, provided that such sensors can fit within the constrained payload dimensions and weight limitations. Moreover, there are plans to integrate sensors to measure carbon dioxide concentrations in the near future.

## 2.5 Flow Simulation

The design of the CopterSonde was mostly driven by the meteorological sampling needs, one of them being the acquisition of
200 data characteristic of the environment. Therefore, a thorough study of the airflow around the CopterSonde and its payload was conducted, with the goal being to mitigate undesired modifications to the sampled air resulting from the CopterSonde body.





To address this challenge, a digital 3D model of the CopterSonde body was created and, subsequently, flow simulations were conducted using CFD included in the SolidWorks® program.

First, real observations were conducted with the CopterSonde in windy conditions at the Kessler Atmospheric and Ecological Field Station (KAEFS) in Purcell, Oklahoma, USA, located 30 km southwest of the OU Norman campus. The CopterSonde completed several hovers while oriented into and perpendicular to the wind. The flights were conducted next to the Washington Oklahoma Mesonet 10 m meteorological tower (WASH; Brock et al., 1995; McPherson et al., 2007), which recorded the atmospheric conditions close to the surface. The external temperature of the CopterSonde body and shell were also recorded using an infrared thermometer. The observations were then used as initial conditions for the flow simulation. A steady state in the simulation was reached after approximately 0.8 sec of simulated physical time, which took about 8 h of computing time. Figure 7 shows a cross-section view of the airflow's temperature with streamlines denoting the air's path. This helped in identifying the air trajectory, patterns, and sources of heat around the CopterSonde and across the sensors. Noting that the blue color is the environmental temperature, a heat aura can be discerned around the CopterSonde. Despite the hottest location being only 1 K higher than the environmental temperature, it can be seen that the region in the air flow corresponding to the smallest temperature perturbation occurs inside the L–shaped duct. Based on the simulations results, the sensing elements are exposed to the cleanest airflow relative to the CopterSonde's surroundings. However, other environmental factors such as the sun radiation were missing in the simulation, which is also a significant contributor to data contamination as found by Greene et al. (2019). Work is underway to produce a more accurate modeling of the CopterSonde.

## 2.6 Calibration

Every new build of the CopterSonde undergoes a series of calibration and validation steps. First, the CopterSonde's autopilot system is tuned to achieve the most stable flight under a large range of weather conditions. Detecting any undesired oscillation or sluggishness in its behavior in this stage is crucial. If not well done, the errors can potentially propagate and impact wind estimations or produce other serious consequences in performance. Another advantage of the modularity of the CopterSonde's sensor payload is that it can be calibrated as a single system instead of its individual components or sensors. The calibration and validation of the sensor package is described in the following sections.

### 2.6.1 Thermodynamic Payload

For calibration of the thermodynamic payload, the entire ducted fan setup was placed inside a reference chamber operated by the Oklahoma Mesonet instruments laboratory. Chamber temperatures ranged from 10–30°C by increments of 10°C every hour at a constant relative humidity of 50%. For relative humidity, the chamber was held at a constant 25°C while varying in relative humidity for an hour at each level. These levels were 15, 35, 55, 75, and 95%. Throughout both of these procedures, the ducted fan was on to continually aspirate the sensors. Individual linear temperature and relative humidity sensor biases were then obtained by comparing 1 min averaged measurements to the reference chamber's National Institute of Standards and Technology (NIST) traceable sensor temperature and relative humidity and computing their average differences. This is the same procedure as discussed in Greene et al. (2019).





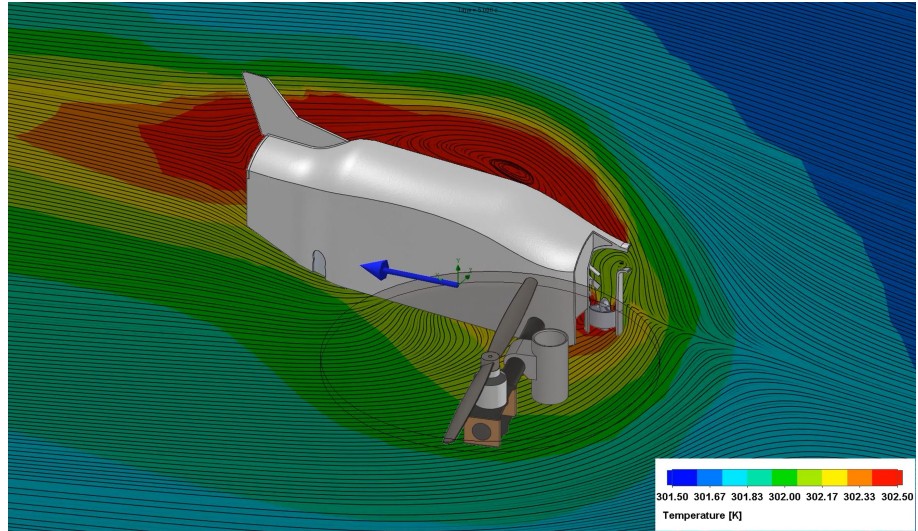

**Figure 7.** Longitudinal cross-section view of the airflow around the CopterSonde CAD model. It shows the CopterSonde facing into winds of $3.6 \text{ m s}^{-1}$ blowing in the direction of the blue arrow. The color spectrum represents the air temperature (in Kelvin) and the streamlines show the flow trajectory. To save computational time and reduce complexity, the rear propellers were removed for the simulation since these are located downwind of the region of interest. Additionally, a section cut view of the L–shaped duct shows the airflow across the sensors and the fan.

235    By applying these calibration offsets, Greene et al. (2019) showed that the CopterSonde temperature measurement spread between the sensors is typically below $0.15°\text{C}$. Bell et al. (2019, in review) compared CopterSonde temperature and relative humidity measurements to radiosondes, and found that CopterSonde temperature measurements are accurate to within $\pm 0.1°\text{C}$ and relative humidity is within $\pm 2\%$.

### 2.6.2    Horizontal Wind Field

240    As mentioned in Sect. 2.3, the CopterSonde is capable of internally making estimations of the horizontal wind field using the kinematic model outlined by Neumann and Bartholmai (2015) and Palomaki et al. (2017). This model is based on a multicopter maintaining its horizontal position tilting into any oncoming wind to divert its vertical thrust vector, which can then be characterized as a function of wind speed. Based on the local coordinate system of the aircraft, the inclination angle (vertical tilt) $\gamma$ is calculated as:

245    $\gamma = \arccos\left(\cos\theta\cos\phi\right),$ (3)

where $\theta$ and $\phi$ are again the pitch and roll angles, respectively. Note, $\gamma$ here is equivalent to $\psi$ defined by Neumann and Bartholmai (2015), but is changed due to the convention of $\psi$ as the aircraft yaw angle. The inclination angle is directly proportional to the drag force $F_d$ by the wind on the aircraft as $\tan\gamma$, and $F_d$ in turn is related to the wind speed encountered





by the multicopter $v$ as:

$$v = \sqrt{\frac{2F_d}{\rho A_{proj} c_d}}, \tag{4}$$

where $\rho$ is the air density, $A_{proj}$ is the aircraft surface area normal to the wind, and $c_d$ is a drag coefficient (Neumann and Bartholmai, 2015). Because of continually changing conditions and orientations of the multicopter, $A_{proj}$ and $c_d$ are generally not well-defined. However, it is possible to estimate wind speed through a linear regression model as:

$$v = C_0 + C_1 \sqrt{\tan\gamma}, \tag{5}$$

where constants $C_0$ and $C_1$ are derived empirically using a common reference measuring wind speed. This model therefore accounts for the intricate design geometry of the aircraft to a first order approximation without the need for complicated expressions for drag coefficients and surface areas.

Environmental wind direction can also be estimated based on the direction of the aircraft tilt. By also incorporating the aircraft heading relative to true North (yaw angle, $\psi$), the wind direction can be evaluated though the same procedure in Sect. 2.3 using Eq. 2.

The procedure for determining coefficients $C_0$ and $C_1$ in Eq. 5 and evaluating the performance of these models for the CopterSonde is nearly identical to that from Sect. 3.1.2 of Greene (2018). The Washington site of the Oklahoma Mesonet is again used as a reference for wind speed and direction, which outputs 1 min data from an RM Young Wind Monitor. Note, the Oklahoma Mesonet nominally produces wind data every 5 min, but we were able to access a data stream from the tower with a better time resolution. The CopterSonde is flown at a hover at 10 m near the tower for 10–15 min at a time on several different days to capture a representative range of conditions for the statistical model. The aircraft inclination angles are then averaged to 1 min intervals to be consistent with the Mesonet data. Linear regression is then performed with the input of $\sqrt{\tan\gamma}$ and Mesonet wind speed as a reference to get a transfer function of the form in Eq. 5. In practice compared to other sampling methodologies like radiosondes and Doppler wind lidars, this method of estimating wind speed with the CopterSonde is accurate to $\pm 0.6$ m s$^{-1}$ from recent calibrations (Bell et al., 2019, in review). Work is underway to improve the wind estimation algorithm, which should reduce the uncertainty.

Similarly, the wind direction estimated by the CopterSonde during these flights is compared to the reference Mesonet tower. Generally, this is a more direct approach and does not require any statistical modeling, unless a constant offset bias is observed. The output angle for the CopterSonde as calculated from Eq. 2 and accounting for the yaw angle $\psi$ tends to be within 4 degrees of the reference in recent calibrations, which is then left uncorrected (Bell et al., 2019, in review). A more in-depth analysis of this methodology for an older model of the CopterSonde is provided in Greene (2018).

## 3   CopterSonde Operations

The CopterSonde system, in its present version and in compliance with our current FAA and OU operating requirements, requires a ground crew of three people for flight operations. The head of operations is the pilot in command (PIC) who is in



charge of the risk assessment and is fully responsible of every decision taken during the operation. The second person is the UAS operator and has the duty of preparing and controlling the CopterSonde by means of a remote control (RC) as needed. The CopterSonde can fly autonomously from takeoff to landing; however, the UAS operator can assume control at any time if necessary. The third person is the GCS operator, who builds the autonomous mission and monitors the CopterSonde's streamed parameters through a computer.

The takeoff of the CopterSonde is usually performed autonomously and initialized by the UAS operator with the authorization of the PIC. The CopterSonde immediately proceeds to execute the pre-loaded waypoint mission. The waypoint mission is the desired flight route for the CopterSonde to follow which is planned and created before the flight by the GCS operator. The CopterSonde is capable of performing a variety of flight routes from simple straight lines to complex spline curves. However, the direct vertical ascent and descent pattern is preferred for temperature and humidity profiling, for which the CopterSonde was
designed and optimized. Letting the CopterSonde land autonomously is safe under good GPS conditions which is necessary for a precise landing, otherwise the UAS operator can take control and manually land the CopterSonde.

### 3.1 Ground Control Station Software

The CopterSonde concept extends beyond the aircraft itself and also includes data handling and distribution systems for the end user. There are three primary components related to the distribution of sensor data from the CopterSonde. The first component
is the sensor package within the flight controller that is responsible for collecting the measurement data, logging to a local log file and transmitting the collected data to the GCS. Sensor data are packed into a custom MAVLink message and transmitted through a telemetry radio to the GCS using ArduPilot's standard messaging platform. The second component is the GCS software, which processes the MAVLink message, plots the live data on various graphical interfaces for visual feedback and forwards the data to the online cloud service. The final component is the data processing systems located within the cloud
computing service Microsoft® Azure. Sensor data are transmitted via the Advanced Message Queueing Protocol (AMQP) to a pre-processing Azure Service Bus, where it then waits to be dequeued by one of the data processing virtual machines. The virtual machines analyze the ingested data, store the results in a cloud database, and finally pass the data to a post-processing Service Bus for interested subscribers.

### 3.2 Deliverables

The CopterSonde system stores the collected data in multiple ways, which is convenient for creating backups and retrieving the data in any circumstance. While in flight, the CopterSonde is continuously broadcasting data through a telemetry radio to the the GCS which stores and uploads the data to the Azure cloud. The processed data are transferred directly from the post-processing Service Bus to the subscribers, as mentioned in Sect. 3.1. For example, the public web application wxuas.com is subscribed to such service and it was created to distribute the CopterSonde's data. The web application then forwards the
information directly to connected web clients via web sockets. This enables a live view of the CopterSonde's data for any remote user via the web during flight operations. Another method of data storage is by using the included removable micro SD card mounted on the Pixhawk board. This method is preferred over the streamed data for post-processing because it stores





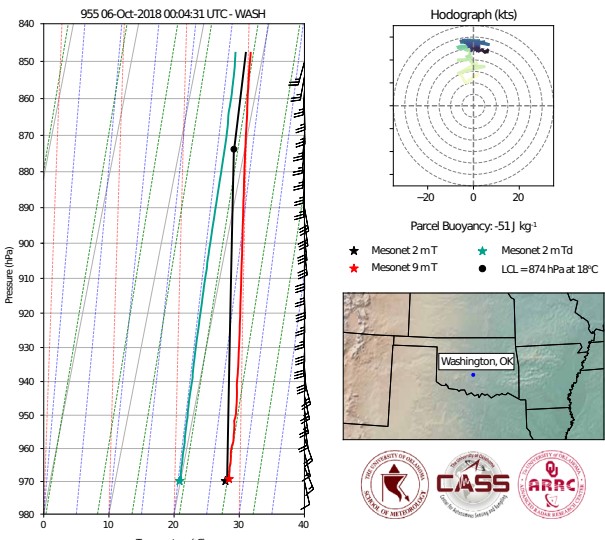

**Figure 8.** Skew–t log–p plot of temperature (red line) and dewpoint (aqua line) generated from a single CopterSonde flight. The surface-based parcel process curve is also included (black line). A hodograph in the upper right corner shows the wind speed and direction over height as well as some basic information from the nearest Oklahoma Mesonet tower. Map is generated using the open-source Basemap package, which references the Generic Mapping Tools dataset (Wessel et al., 2013) for state borders.

higher temporal resolution data. However, retrieving the data from the SD card requires partially taking the CopterSonde apart which hinders the goal of making the system autonomous. Therefore, there are plans to improve data transmission and increase

the temporal resolution of the streamed data to reduce dependence on the SD card transfer. Some useful products generated based on data acquired by the CopterSonde are the Skew–t plots and time–height temperature contours shown in Fig. 8 and Fig. 9 respectively. Post-processing and visualization of CopterSonde data are performed in Python using the open-source packages NumPy (van der Walt et al., 2011), Matplotlib (Hunter, 2007), MetPy (May et al., 2008 - 2019), and SciPy (Virtanen et al., 2019).

**4 Applications**

To date, the a fleet of three CopterSondes have been deployed in several measurement campaigns that have encompassed a wide range of meteorological conditions. Its first appearance was in the Environmental Profiling and Initiation of Convection (EPIC) field project in May 2017 to determine the potential of severe weather development (Koch et al., 2018). Also, during a four weeks field campaign in Hailuoto, Finland, in February 2018, the CopterSonde was operated for the first time in polar

conditions. Known as the Innovative Strategies for Observations in the Arctic Atmospheric Boundary Layer (ISOBAR), it was dedicated to the investigation of the atmospheric boundary layer in the arctic environment (Kral et al., 2018). Surface temperatures dropped to around −20 °C during the campaign with no notable effects in the CopterSonde performance. It





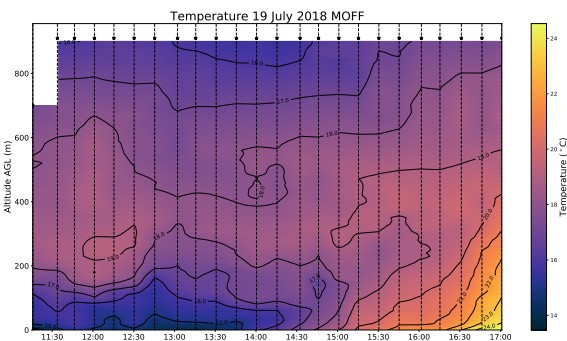

**Figure 9.** Plot of the temperature evolution over time with contour lines up to a height of 914 m. This is the result of interpolating vertical temperature profiles sampled by the CopterSonde every 15 min indicated by the vertical dash lines.

was during this campaign that the CopterSonde achieved its flight altitude record of 1800 m above ground level (AGL). Up to then, the CopterSonde was used to sample the atmosphere for direct scientific evaluations and to assess its limits under
different conditions. Thorough comparisons of the CopterSonde against conventional weather tools occurred later in 2018. The CopterSonde was part of the Lower Atmospheric Profiling Studies at Elevation – A Remotely-piloted Aircraft Team Experiment (LAPSE–RATE) campaign in June 2018, and an in-house Flux Capacitor field campaign in October 2018, during both of which extensive studies of the CopterSonde's performance and data quality were conducted. Comparisons of the CopterSonde's data against data from meteorological towers, radiosondes and a Lidar are discussed in Bell et al. (2019, in
review). A brief overview of cases from the last two field campaigns are provided below.

### 4.1 Case Study 1: LAPSE–RATE

The LAPSE–RATE campaign took place in the San Luis Valley in south-central Colorado. This campaign aimed to study various atmospheric phenomena, such as the morning transition, convection, and drainage flows, using spatially distributed weather sensing UAS from different research institutes (de Boer et al., 2019, accepted). Between the three CopterSonde platforms flown
during LAPSE–RATE, there were over 200 successful flights over a span of five consecutive days. CASS flew at two different locations each day, one of them situated at the Moffat Consolidated School where the Collaborative Lower Atmospheric Mobile Profiling System (CLAMPS, Wagner et al., 2019) was running and launching radiosondes every couple of hours. CLAMPS contains an Atmospheric Emitted Radiance Interferometer (AERI), a HATPRO Microwave Radiometer (MWR), and a Halo Photonics scanning Doppler wind lidar. The AERI and MWR data were combined and run through the AERI optimal esti-
mation (AERIoe Turner and Blumberg, 2018) retrieval to get temperature and moisture profiles of the boundary layer. The Doppler lidar performed Velocity Azimuth Display (VAD) scans and vertical stares. CLAMPS also has the ability to store helium tanks and launch radiosondes.

The CASS team operated under a Certificate of Authorization (COA) granted by the FAA that allowed flights to a maximum altitude of 914 m AGL. The CopterSonde flew vertical profile flights with an ascent rate of $3.5 \text{ m s}^{-1}$ and a decent rate of



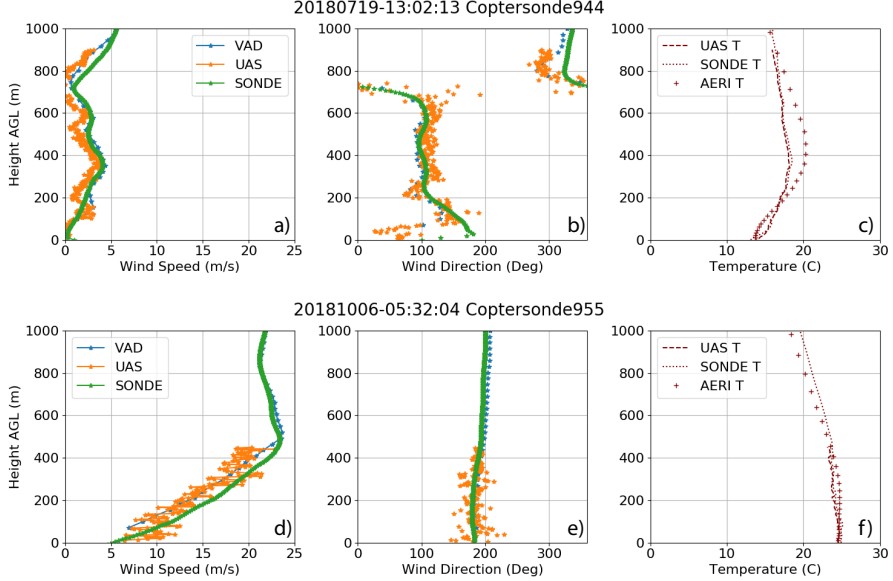

**Figure 10.** Profiles of wind speed (m s$^{-1}$, a and d), wind direction (degrees, b and e), and temperature (°C, c and d). Data from LAPSE–RATE (a-c) are from June 19, 2018 at 13:02 UTC while data from Flux Capacitor (d-e) are from October 6, 2018 at 5:32 UTC.

6 m s$^{-1}$. It took approximately 8 minutes to perform a profile, allowing a 15 minutes cadence for many of the flights. The cadence was determined by the meteorological conditions observed from the live data readout discussed in Sect. 3.1.

Sample data from the Moffat site is shown in Fig. 10(a)-(c) compared to both co-located radiosondes and remote sensors from CLAMPS. During this time period, winds were less than 5 m s$^{-1}$ throughout the CopterSonde profile (Fig. 10a). The CopterSonde generally estimated the wind speed to be approximately 2 m s$^{-1}$ less than both the radiosonde and the Velocity

Azimuth Display (VAD) from the CLAMPS Doppler lidar. The CopterSonde did successfully capture a directional shear layer around 750 m (Fig. 10(b)). Though the wind speed bias falls outside the stated accuracy above ($\pm 0.6$ m s$^{-1}$), it is a consistent bias and can be corrected. Work is ongoing to determine the a calibration procedure for calculating the coefficients in Eq. 5 for an *ascending* rotary-wing UAS, as opposed to a hovering rotary-wing UAS (see Sect. 2.6.2). For this case study, the temperature measured by the CopterSonde is nearly identical to the radiosonde temperature (Fig. 10(c)).

**4.2    Case Study 2: Flux Capacitor**

In the months following LAPSE–RATE, a local, OU coordinated campaign, known as Flux Capacitor, was organized to evaluate the CopterSonde over a full 24 h diurnal cycle. This campaign was held at KAEFS. Similar to LAPSE–RATE, CLAMPS was deployed and radiosondes were launched while flights with the CopterSonde were conducted. During this campaign, OU operated under a COA that allowed flights to a maximum altitude of 1500 m, though the team usually could no longer see the





CopterSonde past approximately 1200 m, and thus was forced to descend to comply with the FAA rule on maintaining the CopterSonde in sight. Fig. 10(d)-(f) shows example data from this campaign.

     This time period was characterized by a strong nocturnal low-level jet (Fig. 10(d)) with wind speeds approaching 25 m s$^{-1}$ at around 500 m. Due to safety concerns, the CopterSonde was commanded to descend if the pitch was greater than 30 degrees due to the high winds. This generally occurred when the wind speed approached 22 m s$^{-1}$. Again, the CopterSonde estimated

wind speeds to be slightly lower (still 2 m s$^{-1}$) than what was measured by the radiosonde (Fig. 10(d)). As mentioned before, this is likely due to the coefficients in Eq. 5 used for the wind estimation. Additionally, the temperature measured by the CopterSonde agree with the observations from the radiosonde (Fig. 10(f)).

     More detailed statistical analyses of how the CopterSonde performs compared to radiosondes and remote sensors can be found in Bell et al. (2019, in review).

## 5    Conclusions


The CopterSonde has been under development for almost two years, during which it time it has undergone near continuous modifications resulting from lessons learned during several successful deployments in diverse locations and under a variety of weather conditions. The CopterSonde is now capable of providing atmospheric measurements of comparable or better quality than many conventional meteorological instruments. For example, it has been shown that the CopterSonde produces

similar results as radiosondes, with the difference being that the CopterSonde is able to control its trajectory and, hence, its sampling location. However, there are still some minor aspects that need to be improved, such as better characterization of the CopterSonde for accurate 2D wind estimations. Despite this, the CopterSonde has been shown to be capable of providing robust and reliable vertical soundings of the ABL. Re-usability and safe operability are also a distinctive feature of the CopterSonde, which makes it flexible to operate in most locations with low operating costs.

The CopterSonde incorporates adaptive atmospheric sampling through its ability to modify a given flight plan based on the sensed environmental parameters, as shown with the wind vane feature. Adding new capabilities and functionalities is a simple process of coding for the autopilot and then conducting initial testing with the ground station simulator before conducting actual flights with the CopterSonde in the field. Moreover, the CopterSonde's adaptive sampling capabilities can provide opportunity to sample the atmosphere in unique ways. For example, the CopterSonde can rapidly determine the temperature gradient while

in flight and, subsequently, adjust its ascent/descent speed to improve the spatial and temporal sampling resolution. Such a feature is currently under development and will be part of the next CopterSonde iteration. It is the intention of the authors to keep maintain an open-source code approach.

     Finally, the real-time data processing and dissemination feature developed for the CopterSonde is a key component for the meteorological community. The rapid availability of high spatial and temporal resolution data can be assimilated into numerical

weather prediction models and, therefore, help to improve the forecasters' situational awareness of prevailing conditions. The authors in collaboration with various modeling teams have begun exploring means to standardize weather data collected by UAS for its assimilation into numerical weather prediction models.



*Code availability.* http://doi.org/10.5281/zenodo.3494656

*Data availability.* Data are available upon request to the corresponding author.

*Competing interests.* The authors declare that they have no conflicts of interest.

*Acknowledgements.* This research has been supported in part by the National Science Foundation under Grant No. 1539070 and internal funding from the University of Oklahoma.



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
