# Peer review of "The CopterSonde: An Insight into the Development of a Smart UAS for Atmospheric Boundary Layer Research"

_Atmospheric Measurement Techniques, 2019_

## Referee Comment (RC1) · Anonymous Referee #2 · 11 Feb 2020

In the manuscript "The CopterSonde: An Insight into the Development of a Smart UAS for Atmospheric Boundary Layer Research" by A. R. Segales et al., the authors introduce the CopterSonde which has the ability to enhance scientific understanding of boundary layer processes by providing routine measurements of temperature, humidity, and wind. The authors describe the CopterSonde's autopilot system and how winds are derived from the platform, provide details on the calibration procedures, and present examples of its measurements from recent case studies. Overall, I find the manuscript to be well-written, and I am happy to see that the CopterSonde is capable of providing robust thermodynamic information. However, I have a few concerns, including about the CopterSonde's winds, which I discuss in more detail below. These

concerns and others need to be addressed before I can recommend publication in Atmospheric Measurement Techniques: 1. I would like to see a graph or two showing the calibration procedure used to determine the coefficients C0 and C1 when operating the CopterSonde next to a tower. What do you mean by a "representative range of conditions"? Does this include very low / nearly calm winds as well? 2. A couple of the figures need to be regenerated. Can you find a better picture to use for the inset images shown in Figure 2? In Figure 8, the CopterSonde's measurements of temperature and dew point temperature were virtually indistinguishable from the rest of the figure. The hodograph also needs to be made larger in order to better distinguish the different features. 3. It is encouraging that the CopterSonde was operated at temperatures down to -20 °C during the ISOBAR field campaign without any significant negative impacts on performance. However, it is unclear what the purpose of Figure 9 is. What do we learn from a simple time-height plot of temperature? Were humidity and winds unavailable from the CopterSonde during ISOBAR? Can you comment on the interpolation procedure used for the temperatures shown here? 4. The CopterSonde shows a significant wind direction bias, particularly for low wind speeds. The CopterSonde's wind directions are more than 100 degrees different from the rawinsonde's observations in the lowest ∼150 m (c.f., Figure 10b). This needs to be addressed. Even around 750-950 m AGL there is also a non-trivial offset, on the order of 45 degrees, between the CopterSonde and rawinsonde's wind directions. 5. Given the rich dataset available from the LAPSE-RATE field campaign, it would be helpful to show additional comparisons between the CopterSonde and other platforms to provide more fidelity in the wind speed and direction measurements obtained from the CopterSonde. How representative were the results from the one case shown in Figure 10a-10c? I suggest showing mean difference plots, with estimates of error, across a range of conditions from the LAPSE-RATE field campaign.

Minor comments: 1. Line 128: "achieved" misspelled 2. Line 349: "descent" misspelled

---

## Referee Comment (RC2) · Anonymous Referee #1 · 20 Feb 2020

General comments:

The manuscript describes the development and application of the CopterSonde as a novel and cost-efficient unmanned system for atmospheric boundary layer research. It is in general well written and provides thorough overview on the technical solution, as well as glimpse on the potential of the system by shortly presenting two case studies. The topic and the overall presentation are well suited for AMT, I have, however identified a few shortcomings that have to be addressed before publishing.

The first paragraph of the introduction is in my opinion rather thin and should be deepened and extended, last but not least, also with a few references. As it stands now

it provides a bit of everything but appears as rather unstructured and low in content. Here the meteorology guys in the group should be able to give some more substantial input. Also the terminology used is in some occasions rather unspecific/unusual (e.g. "numerical modeling simulations")

From a language/grammar point of view, I suggest a thorough overhaul with respect to the use (or mostly non-use) of commas. I feel that there are a lot missing, but I am for sure not the right person to put them all in correctly! But it will improve the readability of numerous passages with rather lengthy sentences considerably.

My main point to address is a certain inconsistency in the description and use of the apparently two different wind speed determination modes, either using the wind vane mode, or not. The description of the wind vane mode and the estimation of the horizontal wind are at the moment not well linked in the manuscript. The wind estimation should be in general much better when performed using the wind vane mode. From the description given, it is not clear if you normally support your wind estimation by the wind vane mode. The description of the wind vane mode in lines 119-123 gives the impression that the wind vane mode is only motivated by the goal of optimizing the flow onto the thermodynamic sensors. But in line 138-139 you outline the motivation for the wind estimation, this should also be mentioned before.

The description of the wind vane mode algorithm in section 2.3 becomes very technical. In my eyes it could in addition also be illustrated/described in a more descriptive way as: the magnitude of the role angle of the CopterSonde is minimized or kept around zero by changing the yaw angle of the vehicle. To assure head wind the yaw is changed in the direction that the pitch is minimized (negative pitch, nose downward).

In summary I highly suggest the manuscript for publication in AMT, after the authors have incorporated my comments.

Specific comments:

Abstract line 13: I suggest to replace "to have" by "to offer" or "to provide"

Line 27: I suggest not to introduce another abbreviation in the UAS/UAV/RPAS/drone world; let's keep it Âńunmanned aerial systemÂż

Figure 1: I suggest to split this figure into two; The Coptersonde deserves one solo picture here in the beginning, maybe something also visually indicating its size; the picture in flight side by side with the mast would fit perectly where you describe your comparison/calibration procedures.

Line 43: replace "in the flight site" by "at the flight site"

Figure 2: should appear bigger in the final manuscript

Line 128: typo in "achievide"

Line 139: add "an" before "additional"

Table 2: the footnotes indicate that the accuracy depends on temperature, but not the response time; I am rather sure that, in particular the response time of the humidity sensor is strongly dependent on temperature!

Figure 5: should appear bigger in the final manuscript

Line 217: replace "sun radiation" by "solar radiation"

Figure 8: hard to read; use the whole page width!

Line 240: "horizontal wind field" should be replaced by "horizontal wind vector". A field consists of several vectors at different locations observed at the same time. The CopterSonde observes a 2D wind vector along a trajectory, one point at a time.

Line 245(Equation 3): Equation 3 can be simplified when applying the wind vane mode and assuming phi=0.

Line 254 (Equation 5): The equation is becoming problematic for asymmetric airframes. A_proj and c_d are very likely to be different for head wind and cross wind. This is

where the wind vane mode becomes important, since the cross wind direction can be neglected and the calibration effort becomes much smaller. This also links to my general comments.

Line 266: "several different days" is a bit confusing; either "several" or "different" alone should be enough; if you use different, you might indicate how many

Line 290: a bit philosophical question, but it is "automatic" or "autonomous" landing? In my opinion it is the first!

Line 290/291: you state you need good GPS for a precise landing; what about the lidar range finder and the IR precision landing camera you have introduced in the beginning?

Line 321: remove "the" after "To date, "

Line 321: replace "have been deployed" by "has been deployed"

Figure 9: should appear bigger in the final manuscript; in particular the labels have to be increased

Figure 10: the units in the caption are not necessary but changing C to degC ($^\circ C$) in c) and f) would look better.

Line 362: has KAEFS been introduced/defined before?

Line 376: replace "during which it time it has" by "during which time it has"

References:

Partially incomplete, in particular volume, issue, page numbering (e.g. Bonin et al. 2015)

Journal names abbreviated (e.g. Brock et al., 1995; McPherson et al. 2005/non-abbreviated

---

## Author Comment (AC1) · 17 Mar 2020

The authors would like to thank the reviewer for their insightful questions and feedback. Included in the supplement are responses to their comments and a revised version of the manuscript detailing changes generated with Latexdiff.

Please also note the supplement to this comment: https://www.atmos-meas-tech-discuss.net/amt-2019-421/amt-2019-421-AC1-supplement.zip

---

## Author Response (AR1)

The authors would like to thank the reviewers and editors for their insightful questions and feedback. These comments have undoubtedly improved the quality of this manuscript. Author responses to each individual comment are outlined below.

Author's response to Anonymous Referee #1
Referee's comments are bold and italicized, and the author's are plain text.

***General Comments:***
***The manuscript describes the development and application of the CopterSonde as a novel and cost-efficient unmanned system for atmospheric boundary layer research. It is in general well written and provides thorough overview on the technical solution, as well as glimpse on the potential of the system by shortly presenting two case studies. The topic and the overall presentation are well suited for AMT, I have, however identified a few shortcomings that have to be addressed before publishing.***
***The first paragraph of the introduction is in my opinion rather thin and should be deepened and extended, last but not least, also with a few references. As it stands now it provides a bit of everything but appears as rather unstructured and low in content. Here the meteorology guys in the group should be able to give some more substantial input. Also the terminology used is in some occasions rather unspecific/unusual (e.g. "numerical modeling simulations").***

We have expanded the introduction with additional references per the reviewer's request.

The first 2 paragraphs of the introduction now reads:

The atmospheric boundary layer (ABL) is a dynamic system that experiences significant changes in the thermodynamic and kinematic states in its vertical and horizontal structure. An understanding of these structures is key for improving numerical modeling, simulations, and weather forecasts. A combination of fine-scale domain models along with higher resolution observations in space and time are required in order to advance such understanding. The measurement of temperature, humidity, pressure and winds are some of the most important parameters for the description of the thermodynamic and kinematic behavior of the ABL. There are currently several meteorological instruments able to measure these parameters effectively; however, they are limited in coverage and have high operating costs. This has resulted in a ``data gap'' in the ABL, which is gaining national and international attention (National Research Council, 2009; Hardesty and Hoff, 2012; Geerts et al., 2017; National Academies of Sciences, Engineering, and Medicine, 2018).

The National Academies has initiated and overseen two ``Decadal Surveys'' over the last 20 years with the goal of generating ``recommendations from the environmental monitoring and Earth science and applications communities for an integrated and sustainable approach to the conduct of the U.S. government's civilian space-based Earth-system science programs.'' (National Academies of Sciences, Engineering, and Medicine, 2018). The 2017-2027 Decadal Survey, released in January 2018, states ``Earth science and derived Earth information have become an integral component of our daily lives, our business successes, and society's capacity to thrive. Extending this societal progress requires that we focus on understanding and

reliably predicting the many ways our planet is changing." (National Academies of Sciences, Engineering, and Medicine, 2018).

*From a language/grammar point of view, I suggest a thorough overhaul with respect to the use (or mostly non-use) of commas. I feel that there are a lot missing, but I am for sure not the right person to put them all in correctly! But it will improve the readability of numerous passages with rather lengthy sentences considerably.*

Thorough proofreading and editing were made. See attached supplement for further details of the changes in the text.

*My main point to address is a certain inconsistency in the description and use of the apparently two different wind speed determination modes, either using the wind vane mode, or not. The description of the wind vane mode and the estimation of the horizontal wind are at the moment not well linked in the manuscript. The wind estimation should be in general much better when performed using the wind vane mode. From the description given, it is not clear if you normally support your wind estimation by the wind vane mode. The description of the wind vane mode in lines 119-123 gives the impression that the wind vane mode is only motivated by the goal of optimizing the flow onto the thermodynamic sensors. But in line 138-139 you outline the motivation for the wind estimation, this should also be mentioned before.*

*The description of the wind vane mode algorithm in section 2.3 becomes very technical. In my eyes it could in addition also be illustrated/described in a more descriptive way as: the magnitude of the role angle of the CopterSonde is minimized or kept around zero by changing the yaw angle of the vehicle. To assure head wind the yaw is changed in the direction that the pitch is minimized (negative pitch, nose downward).*

The first 3 paragraphs of section 2.3 have been reworded as suggested to better describe the wind vane mode as well as to explain how it is linked with the wind vector post-processing.

*In summary I highly suggest the manuscript for publication in AMT, after the authors have incorporated my comments.*

*Specific comments:*
*Abstract line 13: I suggest to replace "to have" by "to offer" or "to provide"*
Fixed. Replaced "to have" with "to provide".

*Line 27: I suggest not to introduce another abbreviation in the UAS/UAV/RPAS/drone world; let's keep it "unmanned aerial system"*
Agreed. UAS now stands for Unmanned Aerial System.

*Figure 1: I suggest to split this figure into two; The Coptersonde deserves one solo picture here in the beginning, maybe something also visually indicating its size; the picture in flight side by side with the mast would fit perectly where you describe your comparison/calibration procedures.*

We agree with this. Figure 1 was replaced with a solo picture of the CopterSonde. Also, an action picture of the CopterSonde by the Oklahoma Mesonet tower (Figure 8) was added in Section 2.6.2 (Horizontal Wind Vector).

**Line 43: replace "in the flight site" by "at the flight site"**
Fixed.

**Figure 2: should appear bigger in the final manuscript**
Figure 2 was replaced with a side view picture of the CopterSonde as per suggestion of Referee #2. The number of inset pictures was reduced to one. The new figure shows the locations of the Lidar and IrLock camera more clearly.

**Line 128: typo in "achievide"**
Fixed.

**Line 139: add "an" before "additional"**
Fixed.

**Table 2: the footnotes indicate that the accuracy depends on temperature, but not the response time; I am rather sure that, in particular the response time of the humidity sensor is strongly dependent on temperature!**
Correct. We've decided to include "according to the manufacturers" in the caption. Additionally, we merged footnotes "b" and "c" and replaced "at 25C temperature" with "Standard Ambient Temperature and Pressure (SATP)".

**Figure 5: should appear bigger in the final manuscript**
Fixed.

**Line 217: replace "sun radiation" by "solar radiation"**
Fixed.

**Figure 8: hard to read; use the whole page width!**
Fixed (now Figure 10).

**Line 240: "horizontal wind field" should be replaced by "horizontal wind vector". A field consists of several vectors at different locations observed at the same time. The CopterSonde observes a 2D wind vector along a trajectory, one point at a time.**
Agree. Replaced wind field with wind vector.

**Line 245(Equation 3): Equation 3 can be simplified when applying the wind vane mode and assuming phi=0.**

The simplification is worth doing when implementing on a microcontroller with low computational resources. However, since the raw data is recorded for later post-processing on a computer, we can account for the relatively small roll angles to determine a best guess of the wind speed.

**Line 254 (Equation 5): The equation is becoming problematic for asymmetric airframes. A_proj and c_d are very likely to be different for head wind and cross wind. This is where the wind vane mode becomes important, since the cross wind direction can be neglected and the calibration effort becomes much smaller. This also links to my general comments.**
Correct. We reworded the motivation of the wind vane mode in Section 2.2 (lines 129-136). Also, a better description of wind vane mode concept was included in the first 3 paragraphs of Section 2.3, before it becomes more technical.

**Line 266: "several different days" is a bit confusing; either "several" or "different" alone should be enough; if you use different, you might indicate how many.**
Fixed. "Different" was removed.

**Line 290: a bit philosophical question, but it is "automatic" or "autonomous" landing? In my opinion it is the first!**
We've decided to keep the word "autonomously" in the specified line of the text. The use of such terms depends on the context. We think it is right to say that the CopterSonde is an automated (or automatic) equipment as a general description, since it operates with little human control. However, if the task is isolated, such as the landing process, the CopterSonde does it autonomously by making its own decisions while attempting to land.

**Line 290/291: you state you need good GPS for a precise landing; what about the lidar range finder and the IR precision landing camera you have introduced in the beginning?**
These text lines were reworded for a better understanding. It now mentions the IR-LOCK precision landing system as an option to increase the precision of the landings.

**Line 321: remove "the" after "To date, "**
Fixed.

**Line 321: replace "have been deployed" by "has been deployed"**
Fixed.

**Figure 9: should appear bigger in the final manuscript; in particular the labels have to be increased**
Fixed (now Figure 11).

**Figure 10: the units in the caption are not necessary but changing C to degC ($^\circ C$) in c) and f) would look better**

Fixed (now Figure 12).

**Line 362: has KAEFS been introduced/defined before?**
Yes. KAEFS was introduced and defined as Kessler Atmospheric and Ecological Field Station in line 225.

**Line 376: replace "during which it time it has" by "during which time it has"**
Fixed.

**References: Partially incomplete, in particular volume, issue, page numbering (e.g. Bonin et al. 2015). Journal names abbreviated (e.g. Brock et al., 1995; McPherson et al. 2005/nonabbreviated**
Fixed.

AMTD Author's response to Anonymous Referee #2
Referee's comments are bold and italicized, and the author's are plain text.

***General Comments: In the manuscript "The CopterSonde: An Insight into the Development of a Smart UAS for Atmospheric Boundary Layer Research" by A. R. Segales et al., the authors introduce the CopterSonde which has the ability to enhance scientific understanding of boundary layer processes by providing routine measurements of temperature, humidity, and wind. The authors describe the CopterSonde's autopilot system and how winds are derived from the platform, provide details on the calibration procedures, and present examples of its measurements from recent case studies. Overall, I find the manuscript to be well-written, and I am happy to see that the CopterSonde is capable of providing robust thermodynamic information. However, I have a few concerns, including about the CopterSonde's winds, which I discuss in more detail below. These concerns and others need to be addressed before I can recommend publication in Atmospheric Measurement Techniques.***

***1. I would like to see a graph or two showing the calibration procedure used to determine the coefficients C0 and C1 when operating the CopterSonde next to a tower. What do you mean by a "representative range of conditions"? Does this include very low / nearly calm winds as well?***

We added figure 9, which is a comparison of CopterSonde calibrated estimated wind speeds versus the reference Oklahoma Mesonet tower. It is then referenced in the text in lines 289--290. We tested wind speeds from 2--11 m/s, which are fairly calm conditions at the low end. See Bell et al. (2020, in review; cited in text) for further discussion on how the wind speed estimations perform.

***2. A couple of the figures need to be regenerated. Can you find a better picture to use for the inset images shown in Figure 2? In Figure 8, the CopterSonde's measurements of temperature and dew point temperature were virtually indistinguishable from the rest of the figure. The hodograph also needs to be made larger in order to better distinguish the different features.***

Figure 2 was replaced with a side view picture of the CopterSonde. The number of inset pictures was reduced to one. The new figure shows the locations of the Lidar and IrLock camera on the CopterSonde more clearly.

Updated figure 8 (now figure 10) to one from the LAPSE-RATE campaign. Line widths were increased for the plotted data and the size of the hodograph was increased.

**3. It is encouraging that the CopterSonde was operated at temperatures down to -20 ◦C during the ISOBAR field campaign without any significant negative impacts on performance. However, it is unclear what the purpose of Figure 9 is. What do we learn from a simple time-height plot of temperature? Were humidity and winds unavailable from the CopterSonde during ISOBAR? Can you comment on the interpolation procedure used for the temperatures shown here?**

Figure 9 (now Figure 11) was generated with data collected during the LAPSE-RATE field campaign. We added context to the caption of figure 9 (now figure 11). It now reads: "Plot of the temperature evolution over time with contour lines up to a height of 914 m. Each CopterSonde profile is separated by about 15 min, and is denoted by vertical dashed lines. The contours and color fill are produced by interpolating each observation level in time, resulting in a rectangular time-height cross-section."

Figure 10 and 11 are just examples of CopterSonde profiles visualisations. These two perspectives enable atmospheric scientists to understand small-scale ABL processes in frameworks they are already familiar with from radiosondes and ground-based remote sensors. Lines 344--345 were added.

**4. The CopterSonde shows a significant wind direction bias, particularly for low wind speeds. The CopterSonde's wind directions are more than 100 degrees different from the rawinsonde's observations in the lowest ~150 m (c.f., Figure 10b). This needs to be addressed. Even around 750- 950 m AGL there is also a non-trivial offset, on the order of 45 degrees, between the CopterSonde and rawinsonde's wind directions.**

We agree with the reviewer. Below 200 m, the wind speeds presented in panels a) an b) are very light, which makes it difficult to reliably estimate the wind direction. In the case of the radiosonde, these data have been averaged in height by the software provided by the manufacturer. Moreover, radiosonde data can be unreliable below 100-200 m because of the 'pendulum effect' created by the swinging movements of the instrument package after release. The agreement between the CopterSonde and VAD wind direction is better (Figure 12 was improved and VAD data is more visible); although, there is still significant scatter, likely because of the low wind. For the data between 750-950 m AGL, we again see an offset in wind direction between the radiosonde and the UAS (and VAD). Here again, the agreement is better between the UAS and VAD. Both the UAS and VAD are true vertical profiles, while the radiosonde will have drifted horizontally with the mean wind. The measurements were collected in a complex terrain and there is likely considerable spatial (horizontal) variability in the wind direction above the region of directional wind shear. Based on the reviewer's comments, we have modified the text to reflect some of these thoughts in the manuscript.

Third paragraph of section 4.2 now reads:

Sample data from the Moffat site is shown in Fig.12a-c compared to both co-located radiosondes and remote sensors from CLAMPS. During this time period, winds were less than 5 m s-1 throughout the CopterSonde profile (Fig.12a). The CopterSonde generally estimated the wind speed to be approximately 2 m s-1 less than both the radiosonde and the Velocity Azimuth Display (VAD) from the CLAMPS Doppler lidar. The CopterSonde did successfully capture a

directional shear layer around 750 m (Fig.12b). It should be noted that the wind direction data presented in Fig.12b reveal better agreement between the CopterSonde and VAD than the CopterSonde and radiosonde. Since the radiosonde drifts with the wind, only the measurements from the CopterSonde and VAD represent true vertical profiles. Moreover, at lower altitudes, below 200 m AGL, there is considerable scatter in the wind direction measurements from the CopterSonde and VAD, likely because of the low speeds. The radiosonde data have been smoothed over height. Here again there is better agreement between the CopterSonde and VAD. The swinging motion of the instrument package on a radiosonde can produce erroneous result at low altitudes after release. Though the wind speed bias falls outside the stated accuracy above (+-0.6m s-1), it is a consistent bias and can be corrected. Work is ongoing to determine the a calibration procedure for calculating the coefficients in Eq.5 for an ascending rotary-wing UAS, as opposed to a hovering rotary-wing UAS (see Sect. 2.6.2). For this case study, the temperature measured by the CopterSonde is nearly identical to the radiosonde temperature (Fig.12c).

*5. Given the rich dataset available from the LAPSE-RATE field campaign, it would be helpful to show additional comparisons between the CopterSonde and other platforms to provide more fidelity in the wind speed and direction measurements obtained from the CopterSonde. How representative were the results from the one case shown in Figure 10a-10c? I suggest showing mean difference plots, with estimates of error, across a range of conditions from the LAPSE-RATE field campaign.*

The paper's scope is mainly about the description of the CopterSonde platform and it's capabilities. Although some examples of case studies were presented, more detailed comparisons with rich dataset 
[revised manuscript text omitted]

---

## Author Response (AR2)

The authors would like to thank the reviewers and editors for their insightful questions and feedback. These comments have undoubtedly improved the quality of this manuscript. Author responses to each individual comment are outlined below.

Author's response to Anonymous Referee #1
Referee's comments are bold and italicized, and the author's are plain text.

***During my final Reading I just found some minor Points:***
***in lines 45, 166 and 169, I would prefer to have a comma upfront "respectively", but this might be an issue that the editor should finally decide on***
Fixed. Commas were added.

***line 113: insert "," after "needed"***
Fixed.

***line 162: insert "the" before "fine-scale"***
Fixed.

***line 235: replace "simulations" by "simulation"***
Fixed.

***line 363: "Lidar" should be lower case "lidar" as in the rest of the manuscript***
Fixed.

***line 406: either "coefficients in use" or "coeficcients used"***
Fixed.

***Reference Koch: journal name missing***
Fixed.

Author's response to Anonymous Referee #2
Referee's comments are bold and italicized, and the author's are plain text.

***In the revised version of the manuscript "The CopterSonde: An Insight into the Development of a Smart UAS for Atmospheric Boundary Layer Research" by A. R. Segales et al., the authors have adequately addressed my concerns about the original version of the manuscript. I have a few minor edits that need to be addressed prior to publication:***
***Line 27: Text cut off***
Fixed.

***Line 165: I suggest rephrasing " the inherent motor wash makes a big limitation" to " the inherent motor wash causes a big limitation"***
Fixed.

***Line 170: Clarify at what frequency the wind vane mode algorithm is calculating the wind vector***
This has been clarified later in the section, last paragraph to be specific.

***Line 329-330: What are "good GPS conditions"?***
A following sentence describing good GPS conditions has been added.

***Line 331: In this context, the word "anytime" is two words***
Fixed.

***Figure 10: Include color scale on hodograph.***
Fixed.

***Line 365: Rephrase "rose new design ideas"***
Fixed.

***Line 394: Change "get" to "obtain"***
Fixed.

[revised manuscript text omitted]